# Genomic Characterization of Carbapenem-Resistant *Klebsiella pneumoniae* ST1440 and *Serratia marcescens* Isolates from a COVID-19 ICU Outbreak in Ecuador

**DOI:** 10.3390/microorganisms13102286

**Published:** 2025-10-01

**Authors:** Estefanía Tisalema-Guanopatín, Fausto Cabezas-Mera, Álvaro A. Pérez-Meza, Veronica Palacios, Franklin Espinosa, Edison Ligña, Ana Cristina Aguilar, Jorge Reyes-Chacón, Michelle Grunauer, Daniel Garzón-Chavez

**Affiliations:** 1Facultad de Medicina y Ciencias, Universidad San Sebastián, Santiago 7510157, Chile; etisalemag@correo.uss.cl; 2Programa de Doctorado en Informática Aplicada a Salud y Medio Ambiente, Universidad Tecnológica Metropolitana, Santiago 8940577, Chile; fcabezasme@utem.cl; 3School of Medicine, Universidad San Francisco de Quito, Quito 170901, Ecuador; aaperezm@usfq.edu.ec (Á.A.P.-M.); aaguilar@usfq.edu.ec (A.C.A.); mgrunauer@usfq.edu.ec (M.G.); 4Laboratory of Microbiology, Hospital IESS Quito Sur, Quito 170812, Ecuador; veronica.palacios@iess.gob.ec (V.P.); jorgereyes83@gmail.com (J.R.-C.); 5School of Medicine, Universidad de las Américas, Quito 170124, Ecuador; franklin.espinosa@udla.edu.ec (F.E.); edison.ligna@udla.edu.ec (E.L.); 6Hospital Vozandes Quito, Quito 170521, Ecuador; 7Facultad de Ciencias Químicas, Universidad Central del Ecuador, Quito 170521, Ecuador; 8Centro de Investigación para la Salud en América Latina (CISeAL), Pontificia Universidad Católica del Ecuador, Quito 170143, Ecuador

**Keywords:** *Klebsiella pneumoniae*, ST1440, *Serratia marcenses*, plasmids, co-selective pressure, clusters

## Abstract

The global rise of antimicrobial resistance (AMR), exacerbated by the COVID-19 pandemic, has led to a surge in infections caused by multidrug-resistant (MDR) bacteria. A key driver of this phenomenon is co-selection, where exposure to one antimicrobial promotes resistance to others via horizontal gene transfer (HGT) mediated by mobile genetic elements (MGEs). Carbapenem-resistant Enterobacteriaceae, known for their genomic plasticity, are particularly worrisome; yet genomic data from Latin America—especially Ecuador—remain scarce. This study investigated four carbapenem-resistant clinical isolates (two *Klebsiella pneumoniae* ST1440 and two *Serratia marcescens*) from tracheal aspirates of three ICU patients during a COVID-19 outbreak at Hospital IESS Quito Sur, Ecuador. Phenotypic profiling and whole-genome sequencing were performed, followed by bioinformatic reconstruction of plasmid content. Nineteen plasmids were identified, carrying 70 resistance-related genes, including antimicrobial resistance genes (ARGs), metal resistance genes (MRGs), integrons, transposons, and insertion sequences. Hierarchical clustering revealed six distinct gene clusters, with several co-localizing ARGs and genes for resistance to disinfectants and heavy metals—suggesting strong co-selective pressure. Conjugative plasmids harboring high-risk elements such as *bla*KPC-2, *qacE*, and *Tn4401* were found in multiple isolates, indicating potential interspecies dissemination. These findings emphasize the importance of plasmid-mediated resistance during the pandemic and highlight the urgent need to enhance genomic surveillance and infection control, particularly in resource-limited healthcare settings.

## 1. Introduction

Antimicrobial resistance (AMR) has become an increasingly complex global health threat, driven by limited therapeutic options and widespread misuse of antibiotics and disinfectants—issues that have intensified during the COVID-19 pandemic [1,2]. These selective pressures have contributed to a global rise in hospital-acquired infections caused by multidrug-resistant (MDR) bacteria [3].

In *Enterobacterales*, β-lactam resistance is mainly mediated by β-lactamases, which are classified into four molecular classes (A–D) according to Ambler’s system [4]. Class A includes extended-spectrum β-lactamases (ESBLs) such as CTX-M, TEM, and SHV, as well as carbapenemases such as KPC-2, capable of hydrolyzing penicillins, cephalosporins, and carbapenems [5]. Class B comprises metallo-β-lactamases (MBLs), including NDM, VIM, and IMP, which require zinc and efficiently degrade carbapenems [6]. Class C encompasses AmpC enzymes, usually chromosomally encoded, that confer resistance to cephalosporins. Finally, Class D consists of OXA-type carbapenemases with variable activity [7]. In addition to β-lactam resistance, *Enterobacterales* often carry determinants for fluoroquinolone resistance. These include plasmid-mediated quinolone resistance (PMQR) genes—such as *qnrA*, *qnrB*, *qnrS*, and *aac (6′)-Ib-cr*—as well as chromosomal mutations in DNA gyrase (*gyrA*, *gyrB*) and topoisomerase IV (*parC*, *pare*) [8]. Resistance to aminoglycosides typically involves enzymatic modification via acetyltransferases (*aac*), nucleotidyltransferases (*ant*), and phosphotransferases (*aph*), with these genes often co-located on plasmids alongside other resistance determinants [9].

*Klebsiella pneumoniae* is one of the most worrisome MDR pathogens, responsible for a wide range of healthcare-associated infections, including pneumonia, bloodstream infections, urinary tract infections, and surgical site infections—especially in immunocompromised patients and those admitted to intensive care units [10]. Its clinical success is attributed to a diverse set of virulence factors, such as capsular polysaccharides, siderophores (e.g., enterobactin and yersiniabactin), adhesins (e.g., fimbriae and mannose-resistant *Klebsiella*-like hemagglutinins), and serum resistance proteins [11]. Globally, high-risk clones such as ST258, ST11, ST147, and ST307 have contributed to the dissemination of carbapenem resistance genes, particularly *bla*KPC-2 and *bla*NDM, via epidemic plasmids [12]. These clones display exceptional genomic plasticity, enabling rapid adaptation to selective pressures and efficient horizontal transfer of resistance determinants [13]. *Serratia marcescens* has become an increasingly relevant nosocomial pathogen, especially in intensive care units, where it is associated with ventilator-associated pneumonia, catheter-related bloodstream infections, and urinary tract infections [14]. This opportunistic bacterium displays intrinsic resistance to ampicillin, first-generation cephalosporins, and polymyxins, mainly due to chromosomal AmpC β-lactamase production and lipopolysaccharide modifications [15]. Additionally, *S. marcescens* can acquire resistance mechanisms such as carbapenemases (e.g., KPC-2, NDM, and OXA-48-like), extended-spectrum β-lactamases (ESBLs), and 16S rRNA methylases that confer high-level aminoglycoside resistance [16]. Genomic studies have revealed distinct *S. marcescens* lineages with diverse resistance profiles, including clones harboring multiple β-lactamases and extensive plasmid content [17].

These bacteria efficiently exchange mobile genetic elements (MGEs) through horizontal gene transfer (HGT), allowing rapid adaptation to stress conditions and promoting the spread of antibiotic resistance genes (ARGs) [18,19]. During the COVID-19 pandemic, studies reported a marked increase in ARG abundance and HGT activity, likely triggered by intensified co-selective pressures [20].

Co-selection occurs when exposure to a single selective agent—such as an antibiotic—also promotes resistance to unrelated agents. This can happen through (i) co-resistance, where multiple resistance genes coexist on the same MGE; (ii) cross-resistance, when a single mechanism confers resistance to different agents; and (iii) co-regulation, involving coordinated expression of resistance genes in response to common stimuli [21].

Despite global concern, research on carbapenem-resistant *Enterobacterales* in Latin America—particularly in Ecuador—remains scarce. Most studies in Ecuador focus solely on phenotypic resistance profiles, lacking detailed molecular characterization to elucidate the underlying genetic mechanisms [22]. Although carbapenem-resistant strains have been reported in clinical and food-related settings, information on clonal lineages, mobile genetic elements, and transmission dynamics is still limited [23,24]. This gap hampers the development of effective surveillance and containment strategies, emphasizing the urgent need for genomic studies at the local level.

The COVID-19 pandemic profoundly disrupted the management of nosocomial infections and contributed to the spread of plasmids in healthcare environments. Emergency responses—such as increased antimicrobial use, healthcare system overload, and intensified disinfection practices—altered infection dynamics and heightened the selective pressures that favor MDR bacterial dissemination [25,26].

In this context, the present study aimed to characterize the genomic features of four carbapenem-resistant clinical isolates—two *K. pneumoniae* ST1440 and two *S. marcescens*—obtained during a localized outbreak in a tertiary hospital in Ecuador. Special emphasis was placed on plasmid-mediated resistance and co-selection dynamics under pandemic conditions.

## 2. Methodology

### 2.1. Sample Collection

This study included four carbapenem-resistant bacterial strains. Inclusion criteria were (1) isolates recovered from adult ICU patients presenting clinical signs of bacterial infection during the COVID-19 pandemic, (2) growth on selective culture media, and (3) confirmed carbapenem resistance through initial phenotypic screening. The strains were obtained from tracheal aspirates collected during routine diagnostics at Hospital IESS Quito Sur (Quito, Ecuador) between August and September 2021, coinciding with the second wave of the pandemic in the country. The isolates were obtained from three patients: two *K. pneumoniae* strains (from patients 1 and 2) and two *S. marcescens* strains (from patients 2 and 3). All patients presented with severe respiratory symptoms, including dyspnea, fever, leukocytosis, changes in tracheal secretions, and the need for invasive mechanical ventilation.

### 2.2. Microbiology Techniques

Bacterial identification and antimicrobial susceptibility testing were performed using standard microbiological techniques. Pure colonies from each isolate were cultured on MacConkey agar and adjusted to a 0.5 McFarland standard for fitness assays, conducted in triplicate. An automated blood culture system (BD BACTEC, BD Diagnostics, Franklin Lakes, NJ, USA) with a fluorescent CO_2_ sensor was used to monitor growth at 10-min intervals, with positive detection set at a threshold of 10 CFU/mL. Genotypic identification was confirmed through whole-genome sequencing and multilocus sequence typing (MLST). Species identity was validated using Average Nucleotide Identity (ANI) analysis.

### 2.3. DNA Extraction and Sequencing

Genomic DNA was extracted using the DNeasy UltraClean Microbial Kit (Qiagen, Hilden, Germany), following the manufacturer’s instructions. DNA concentration and purity were assessed using both a NanoDrop One (Thermo Fisher Scientific, Waltham, MA, USA) and a Qubit 3.0 Fluorometer (Invitrogen, Carlsbad, CA, USA), ensuring A260/A280 ratios between 1.8 and 2.0 and concentrations above 20 ng/μL. DNA integrity was verified by agarose gel electrophoresis. Library preparation was carried out with the NEBNext Ultra II DNA Library Prep Kit (New England Biolabs, Ipswich, MA, USA). Sequencing was performed on the Illumina HiSeq 4000 platform, producing 101 bp paired-end reads with a minimum genome coverage of 100×.

### 2.4. Data Availability and Bioinformatic Analysis

Whole-genome sequencing (WGS) was performed, and raw reads were deposited in the Sequence Reads Archive (SRA) under BioProject PRJNA1051122, with BioSample IDs: SAMN38765470 (27026352), SAMN38765469 (2729958), SAMN38765468 (27026351), and SAMN38765467 (2684096). Additional bioinformatics data and analysis files are available at: https://www.bv-brc.org/workspace/sebas007@patricbrc.org/PROJECT_PLASMIDS. (accessed on 30 July 2024).

A multi-stage bioinformatic pipeline was used to ensure quality control, accurate annotation, and reproducibility. Raw reads were assessed with FastQC v0.12.0 and trimmed using Trimmomatic v0.39. Assemblies were generated with Unicycler v0.5.0 and evaluated using QUAST v5.2.0. Plasmid-associated contigs were identified using MOB-suite v3.1.9. To detect mobile genetic elements (MGEs) and horizontal gene transfer (HGT) events, we used mobileOG-db v1.1.3, BacAnt, Alien Hunter v1.1.0, Phigaro v1.0.1, and PHASTEST. Functional annotation and prophage prediction were included in this analysis. CRISPRCasFinder v1.1.0 was employed to identify CRISPR arrays and associated Cas genes. Resistance genes were annotated using ResFinder v4.5.0, CARD v3.2.9, and staramr v0.10.0, applying thresholds of ≥90% identity and ≥60% coverage. The complete list of detected resistance genes is provided in Appendix A. Genome annotations were completed with Bakta, and additional plasmid-like elements were searched using PLSDB (version 2023_11_03_v2). Data analysis and visualization were conducted using Python v3.12.3 with libraries such as pandas, seaborn, and matplotlib.

## 3. Results

### 3.1. General Features

Patient 2 was found to harbor both carbapenem-resistant *K. pneumoniae* (CrKp) and *S. marcescens* (CrSm), corresponding to isolates 27026351 and 27026352, respectively. In contrast, patient 1 carried only CrKp (isolate 2684096), while patient 3 was colonized by CrSm (isolate 2,729,958). All CrKp strains were identified as sequence type ST1440. Species identity and genomic similarity were confirmed using ANI through the TYGS platform. CrKp and CrSm isolates from patient 2 showed high genomic similarity to those from patients 1 and 3, respectively. However, the CrSm strains differed notably in plasmid content. Growth kinetics assays revealed variability between strains and species, although the differences were not statistically significant. Interestingly, CrSm isolate 27,026,352 exhibited the longest growth time among all isolates, possibly due to the metabolic burden imposed by an atypical plasmid.

### 3.2. Chromosomal Features

We observed that the draft chromosomes of the isolates exhibited both consistent and distinctive characteristics corresponding to their respective species, as shown in Table 1. Our analysis identified 32 genes and point mutations of interest on CrKp chromosomes and 13 on CrSm chromosomes, which were classified into five groups: (i) associated with antibiotic resistance, (ii) expression regulators, (iii) components of membranes, efflux pumps, and permeability, (iv) virulence and adhesion factors, and (v) insertion sequences (IS). Each group displayed unique compositions across the two species. In the first group, predominantly associated with AMR, we identified a diverse array in CrKp, including *fosA5*, *parC*, *gyrA*, *gyrB*, *emrR*, *emrD*, *ArnT*, *eptB*, and *rpsL*. In contrast, CrSm displayed a more limited set, featuring only *aac (6′)-Ic* and multiple copies of *blaSRT-1.* In the second group, focused on expression regulators, CrKp showed a richer genetic landscape, with *UhpT*, *acrR*, *ramR*, *marA*, *baeR*, *H-NS*, and *CRP*, while CrSm contained only *CRP.* This disparity suggests potential differences in regulatory mechanisms between the two species. In the third group, encompassing components of membranes, efflux pumps, and permeability, CrKp possessed a wide range, including *OmpA*, *ompK35*, *ompK36*, *ompK37*, *msbA*, *fieF*, *kdeA*, *KpnF*, *KpnG*, *KpnE*, *KpnH*, and *LptD*. In contrast, CrSm exhibited a distinct set: *smfY*, *ssmE*, *sdeY*, and *sdeB.* In the fourth group, focused on virulence and adhesion factors, we observed distinct genetic profiles for each species. CrKp contained *iutA*, *fimH*, and *mrkA*, while CrSm harbored *smdA*, *fyua*, *irp2*, *sdeA*, and *ybtQ.* These differences contribute to the unique virulent characteristics of each species. In the fifth group, concerning insertion sequences (IS), we found ISKpn1 and ISKpn2 in CrKp, with ISRaq1 and ISRaq2 present in CrSm. These groups may play crucial roles in genetic mobility and adaptability within each species.

Furthermore, our results revealed significant enrichment at both metabolic and functional levels in five key pathways: amino acids and derivatives, cofactors and vitamins, energy generation, xenobiotic degradation and stress response, and membrane transport. This pattern indicated the enrichment of three major families of protein CDS: transporters and permeases for diverse substrates, transcriptional regulators of multiple families, and a wide range of enzymes, including hydrolases, transferases, and oxidoreductases.

Considering the average number of copies per genome, the principal CDS products in the four chromosome drafts were a DNA-binding transcriptional regulator (*LysR* family) and a putative *AraJ* arabinose efflux permease (*MFS* family). The distribution of other products was equitable between species and samples, except for those associated with phage, hypothetical proteins, and diverse transcriptional regulators, as shown in Figure 1. These findings suggest substantial metabolic and functional adaptation.

Moreover, we identified putative HGT regions and prophages. CrKp exhibited more HGT regions (123/121) compared to CrSm (93/87), while CrSm contained more prophage regions (5/6) than CrKp (3). Based on the annotations of mobile orthologue groups, which encompass protein families involved in integration/excision (IE), replication/recombination/repair (RRR), and stability/transfer/defense (STD) and phages, our analysis revealed that CKp possessed more RRR genes (242) than CrSm (188), whereas CrSm harbored more phage-related genes (219) than CrKp (146). Intraspecies comparisons showed no significant differences (Figure 2). Notably, CRISPR/Cas system-associated genes were absent in both species. These findings elucidate the distribution of MGEs in CrKp and CrSm, providing insights into their genomic plasticity.

### 3.3. Plasmid Features

We assembled nineteen plasmids, including fourteen from CrKp isolates and five from CrSm isolates, and designated them as AA275 (2 copies), AA002 (4 copies), AB595 (3 copies), AA119 (2 copies), AA531 (4 copies), AB042 (3 copies), and AD092 (2 copies). We categorized the plasmids into conjugative (AA275, AA002), non-mobilizable (AB595, AA119), and mobilizable (AA531, AB042, AD092) types. We obtained only one AA002 plasmid from the CrSm isolate ID 27026352 from the MDB patient. The AA002 plasmid appeared in all four isolates, while AB595, AB042, and AA531 appeared in three isolates. We detected AA119, AA275, and AD092 exclusively in CrKp isolates. Although the AD092 plasmid carried the *dfrA8* gene, it exhibited a unique configuration unrelated to other plasmids, so we excluded it from certain analyses. The largest plasmids observed were the conjugative plasmids AA275 and AA002, averaging 184,931 bp (*n* = 2) and 90,122 bp (*n* = 4), respectively. AB595 averaged 14,991 bp (*n* = 3), while the remaining plasmids had a mean size of 3264 bp (*n* = 7). Conjugative plasmids exhibited a GC content of 53.8 ± 0.2%, non-mobilizable plasmids 51.0 ± 5.4%, and mobilizable plasmids 51.7 ± 4.9%. AB042 and AA119 displayed the lowest GC content at 47.2 ± 0.1% and 46.4 ± 0.0%, respectively. Replicons detected included IncF in AA275, IncM1 in AA002, and both Col (pHAD28) and Col440I in the remaining plasmids. We predicted a circular topology for all recovered plasmids (Table 2).

The main coding sequences (CDS) products were hypothetical proteins distributed equally among species and samples. On the other hand, we observed a marked difference in transposases that were almost exclusively in CrKp plasmids, as opposed to a wealth of transcriptional regulators in CrSm plasmids (Figure 1). AA275 and AA002 exhibited regions indicative of putative HGT events and prophage regions, as well as a higher number of CDS compared to the other plasmids. A higher proportion of elements in the mobile orthologue groups were associated with integration/excision (IE), comprising 163 (44.7%), followed by transfer elements, which included 105 (28.8%). Elements from various groups were evenly distributed in AA002, AA275, and AB595, whereas AA531 contained exclusively transfer elements, and AA119 featured elements related to stability/transfer/defense (STD) and replication/recombination/repair (RRR) (Table 2 and Figure 2).

### 3.4. Clusters

Clustering analysis excluded AA119 and AA531 due to the absence of genes of interest. Among the examined plasmids, we identified seventy genes, including ARGs, metal resistance genes (MRGs), and MGEs such as transposons, insertion sequences (IS), and integrons. To cluster the identified genes, we conducted an occurrence analysis using the average linkage method and Jaccard dissimilarity distance, with a cut-off value of 0.4. The resulting clustering demonstrated high quality, as evidenced by a Silhouette coefficient of 0.89, total inertia of 246.6, and a Davies-Bouldin index of 0.31 (Figure 3).

Clustering analysis identified six robust clusters, each containing at least three genes. Three of these were smaller clusters: Cluster 1 (*Tn4401*, *blaKPC-2*, *qacE*, and *sul1*), Cluster 2 (*ISVsa3*, *blaTEM-1A*, *rmtG*, *sul2*, and *tetD*), and Cluster 3 (*ISKpn11*, *ISKpn12*, and *aac (3)-Iia*). The larger clusters included Cluster 4 (twelve members of diverse gene types), Cluster 5 (twenty-six members, primarily MRGs with IS and transposons, including the heat resistance-associated *clpK1*), and Cluster 6 (nine members, featuring several *blaTEM* gene family variants and associated Tn801). Additionally, we observed three frequently associated gene pairs: *silC* and *silF*, *v* and *blaCTX-M-12*, and *IS5075* and *merT*. Five genes (*blaSHV-1*, *aadA1*, *qnrB19*, *dfrA8*, and *Integron1*) showed no significant associations with others. For more details, refer to Figure 3 and Figure 4.

## 4. Discussion

The COVID-19 pandemic has significantly impacted the management of nosocomial infections and plasmid dissemination in healthcare settings, particularly in developing countries. Emergency measures to control SARS-CoV-2, including increased antimicrobial usage, healthcare system overload, and enhanced infection control protocols, have disrupted the typical dynamics of nosocomial infections. These changes have intensified the selective pressures that promote the selection and spread of MDR bacteria, promoting HGT, especially of plasmid-borne genes, and exacerbating cross- resistance problems. Understanding these evolutionary dynamics is crucial for developing effective control strategies in resource-limited settings during future pandemics.

In Ecuador, epidemiological studies have focused on isolates of *K. pneumoniae* belonging to the high- risk clonal complex CG258, which includes globally widespread sequence types (STs) such as ST258, ST25, ST348, and ST512 (23,27). Additionally, isolates with unique sequence types, including ST42, ST111, ST147, ST859, ST1088, ST1199, ST1758, and ST1850 [27], have been identified in more localized nosocomial outbreaks. In this study, we describe two ST1440 isolates, a lineage for which limited information is available. This data scarcity is reflected in the specialized database (https://bigsdb.pasteur.fr/klebsiella/) (accessed on 14 October 2024), where only seventeen isolates have been reported, with just three originating from Latin America.

In recent years, *S. marcescens* isolates have harvested significant attention due to their increasing AMR in nosocomial environments and the diversification of their resistance mechanisms, particularly those mediated by plasmids [28]. Matteoli et al., through a comprehensive pangenome analysis, identified twelve distinct lineages, with Sm7 and Sm9 exhibiting the most worrying resistome profiles [17]. Sm7 lineage demonstrated widespread distribution of *bla*KPC-2-harboring plasmids with lower plasmid diversity (one per genome), while Sm9 showed the highest plasmid diversity (averaging three per genome) and multiple beta-lactamases. These findings align with numerous studies indicating that carbapenem resistance in nosocomial isolates is attributed to the presence of plasmid-borne *bla*KPC genes, with a secondary contribution from beta-lactamase overexpression and porin loss [16,29].

Studies have reported varying rates of bacterial superinfection in patients with SARS-CoV-2 infection. Initial reports documented a 16% prevalence [30], while subsequent meta-analyses found superinfections in 24% of COVID-19 patients, correlating with increased mortality risk [31]. Other investigations revealed coinfection rates ranging from 3.5% to 14% [32,33]. An intensive care unit (ICU) study found even higher rates, with respiratory coinfections at 33.3% and superinfections at 43.9%. In this ICU population, the most frequently isolated bacteria were *Pseudomonas aeruginosa*, *Enterococcus faecium*, *K. pneumoniae*, and *Acinetobacter baumannii* [34]. Despite the variable rates of confirmed bacterial infections, Rawson et al. reported widespread use of broad-spectrum antibacterials, with 72% of COVID-19 patients receiving antimicrobial therapy. However, these analyses lacked crucial data on antimicrobial susceptibility patterns and prescribing practices [25]. Collectively, these findings highlight not only the significant variability in respiratory coinfection and superinfection rates among COVID-19 patients across different clinical settings but also raise concerns about potentially excessive antibiotic use.

The pandemic period saw a concerning rise in multidrug-resistant organisms (MDROs), including both bacteria and fungi [35]. i et al. documented particularly high bacterial resistance rates, with *K. pneumoniae* showing 75.5% carbapenem resistance and *A. baumannii* reaching 91.2% resistance [36]. This increase in resistant organisms likely stemmed from multiple factors during the COVID-19 pandemic: hospital units faced overwhelming patient admissions, experienced shortages in personal protective equipment, and increased broad-spectrum antimicrobial usage. These challenges collectively may have contributed to enhanced transmission of MDROs within hospital settings.

The available literature in Ecuador is very scarce [22]. This combined scenario underlines the urgent local need to intensify the study and reporting of these nosocomial strains. During the first stage of the analysis of the draft chromosomes, we identified multiple genes involved in the degradation of xenobiotics such as 1,4-dichlorobenzene, tetrachloroethylene, 2,4-dichlorobenzoate, and others. These are surprising and worrisome. This finding is consistent with several studies that indicate oxidative stress caused by using cleaning products, disinfectants, and solvents as significant drivers of AMR. This oxidative stress can increase HGT by up to 8-fold, resulting in a 20-fold increase in ARGs abundance [20,37]. Moreover, exposure to these chemicals can induce the Viable-But-Non-Culturable (VBNC) state in nosocomial strains [38].

These findings suggest that these strains have adapted their genomes to these pressures to tolerate and metabolize, to some extent, common disinfectants and solvents used in healthcare environments. This phenomenon is probably intensified by the high dynamics of their membranes, which makes them highly selective, either by point mutations in antimicrobial targets or by the expression of multiple efflux pumps. For example, two CRISPR arrays were identified on each chromosome of *K. pneumoniae*, although these sequences serve as Level 1 evidence and may potentially yield false positives [39]. This finding is consistent with several studies collectively indicating an inverse relationship between the presence of CRISPR/Cas systems and other restriction-modification systems and the capacity to acquire MGEs, thereby facilitating their dissemination, such as epidemic IncF plasmids. This phenomenon has been observed in strains ST258, ST307, ESBL-producing strains, strains resistant to carbapenems, and various clinical isolates [40]. Furthermore, sequences associated with type I cas3 were detected on the AA275 plasmids. It has been reported that the presence of Cas3 on ColE1 plasmids can induce uncontrolled replication into concatemers at 37 °C, solely driven by the helicase activity of the enzyme without requiring other CRISPR-Cas components [41].

However, complete CRISPR/Cas systems have been observed in much larger plasmids [42] (>200 kb) with multiple ARGs, associated with replicons such as IncFIB and IncHI1B [42]. Our study reports the presence of Cas3 on non-ColE1 plasmids and the absence of CRISPR arrays in ST1440 strains, potentially indicating an optimal scenario for the plasmid-borne ARGs in nosocomial environments. However, experimental validation is required. Additionally, sequences associated with type I Cas3 were detected in the AA275 plasmids. It has been documented that the presence of Cas3 in ColE1 plasmids can lead to uncontrolled replication in concatemers at 37 °C, driven solely by the helicase activity of the enzyme without the need for other CRISPR-Cas components. This study identifies the presence of Cas3 in plasmids other than ColE1 and the absence of CRISPR arrays in ST1440 strains, suggesting an optimal scenario for the transmission of plasmid-mediated AMR in hospital settings. Our bacterial fitness assay shows a potential high growth rate related to ColE1 plasmids, which could bring an advantage to bacteria that have it; however, metabolic-related pathways with these findings still need to be labeled.

In conclusion, this dual outbreak affecting three patients from the same ward, infected with CrKp and CrSm, underscores the complexity of nosocomial infections caused by enteropathogens. The occurrence of two patients infected with CrSm, two with CrKp, and one case of co-infection by both pathogens presents a concerning scenario. HGT via plasmids shared among different pathogenic *Enterobacteriaceae* species represents a significant threat, particularly when co-selective pressures exist, which could exacerbate AMR in alignment with WHO predictions for 2050. Continuous monitoring of this dynamic is not only crucial for analyzing the efficacy of control measures but also for mitigating the alarming projections detailed in the O’Neill report [43]. This case serves as a valuable model for developing cost-effective strategies to address the growing threat of antimicrobial resistance. Furthermore, it highlights the urgent need for comprehensive surveillance and innovative approaches to combat the spread of MDR organisms in healthcare settings.

## Figures and Tables

**Figure 1 microorganisms-13-02286-f001:**
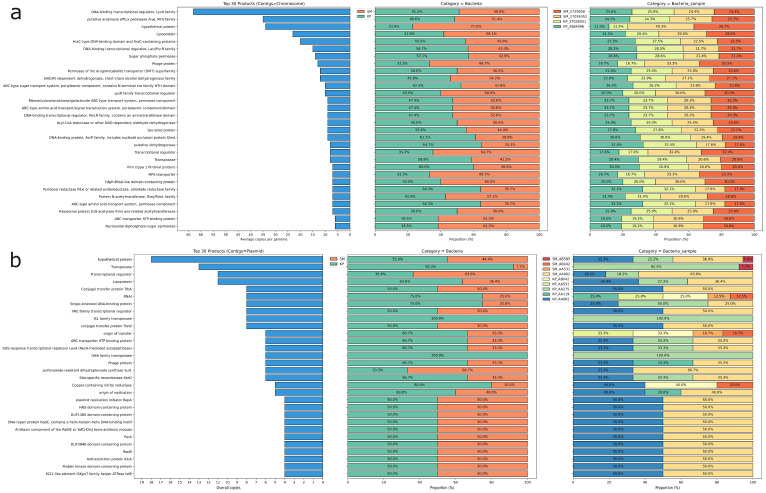
Distribution of predicted protein products in chromosomal (**a**) and plasmid (**b**) contigs. The left-side bar plots show the top 30 predicted protein products for each contig based on the average copy number per genome. The stacked bar plots on the right represent the distribution of these products across species (middle) and specific samples (right). In (**a**), dominant products include DNA-binding transcriptional regulators, lipoproteins, and ABC transporters, while (**b**) highlights plasmid-related proteins such as transposases, transcriptional regulators, and conjugation proteins. The color legend indicates the specific isolates analyzed.

**Figure 2 microorganisms-13-02286-f002:**
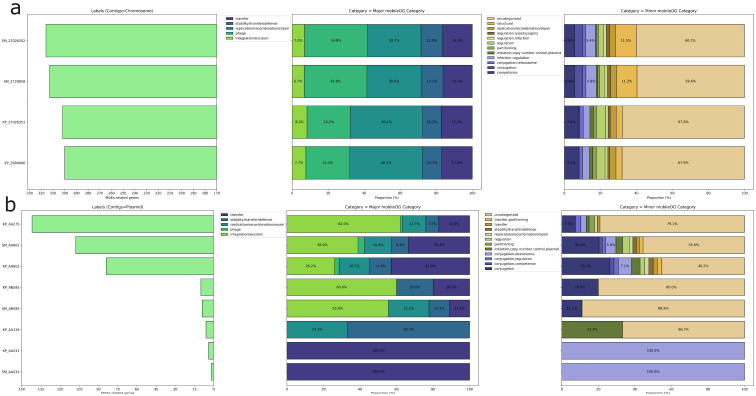
Distribution of mobile genetic element (MGE)-related genes in chromosomal (**a**) and plasmid (**b**) contigs. The left-side bar plots show the total number of MGE-related genes for each contig, while the stacked bar plots on the right represent the distribution of these genes across major and minor mobileOG categories. Major categories include functions such as transfer, replication/recombination/repair, phage, and integration/excision, while minor categories provide more detailed functional classifications, including regulation, conjugation, and plasmid replication control.

**Figure 3 microorganisms-13-02286-f003:**
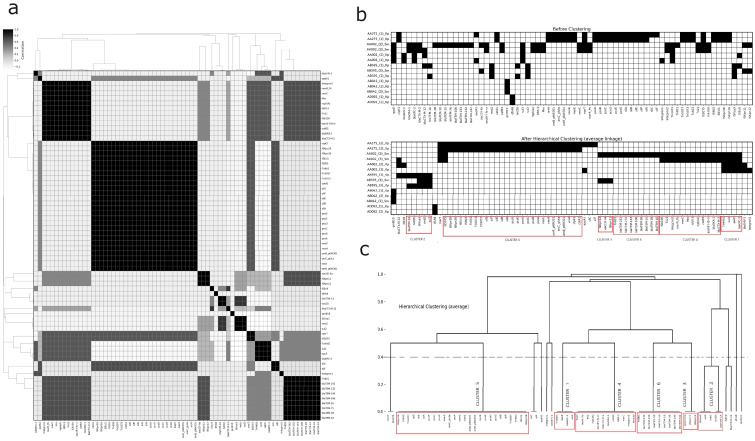
Correlation analysis and hierarchical clustering of plasmid-borne antimicrobial resistance (AMR) gene profiles. (**a**) Heatmap displaying pairwise correlations between AMR genes across different genomes, with correlation values ranging from white (no correlation) to black (high correlation), accompanied by dendrograms representing hierarchical clustering based on gene similarity. (**b**) Presence/absence matrix of AMR genes shown before and after applying hierarchical clustering using the average linkage method, with distinct clusters highlighted in red boxes revealing coherent groups of AMR genes shared across genomes. (**c**) Dendrogram generated from the clustering process, grouping AMR genes into clusters that reflect their genomic similarities, with each cluster clearly labeled and enclosed in red boxes. This comprehensive visualization aids in understanding the organization and relationships of AMR genes across multiple genomes.

**Figure 4 microorganisms-13-02286-f004:**
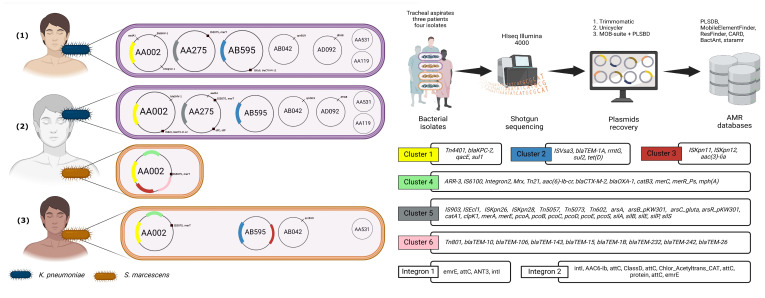
Schematic representation of plasmid and antimicrobial resistance (AMR) gene distribution in clinical isolates obtained from tracheal aspirates of three patients [(1), (2), (3)]. The plasmids recovered from *K. pneumoniae* (blue) and *S. marcescens* (orange) isolates are displayed on the left, with annotated key resistance genes. The middle section illustrates the experimental workflow, starting with bacterial isolate recovery from tracheal aspirates, followed by shotgun sequencing using HiSeq Illumina 4000, and subsequent plasmid reconstruction using bioinformatics tools. AMR gene identification was performed using databases such as PLSDB, ResFinder, CARD, and MobileElementFinder. The right panels detail the six AMR gene clusters and integrons (1 and 2), showing genes related to resistance mechanisms, including *bla*KPC-2, *aac (3)-lia*, *Tn4401*, and various transposons and integron cassettes, revealing the genetic architecture of resistance across the isolates.

**Table 1 microorganisms-13-02286-t001:** Overall characteristics of WGS assemblies, chromosomes, and plasmids.

General Features				
Patient	(1)	(2)	(2)	(3)
SRA accession	SRX22858523	SRX22858524	SRX22858526	SRX22858525
Isolate ID	2684096	27026351	27026352	2729958
Species	*K. pneumoniae*	*K. pneumoniae*	*S. marcescens*	*S. marcescens*
MLST	ST1440	ST1440	-	-
Fitness assay (min ± SD)	208.3 ± 19.9	200.0 ± 10.4	231.0 ± 35.5	206.3 ± 16.7
**Assembly quality**				
Overall length (bp)	5,381,579	5,386,326	5,334,682	5,352,178
Contigs count	61	61	42	45
GC content (%)	57.46	57.46	59.72	59.7
Largest contig (bp)	631,437	631,437	1,802,735	2,510,374
N50 (bp)	292,952	311,324	466,601	356,805
N90 (bp)	56,091	58,962	119,513	119,513
auN (bp)	295,859.8	321,543.1	841,142	1,298,789.8
L50	7	7	3	2
L90	22	20	11	11
**Chromosome**				
Contigs counts	36	35	39	37
Length (bp)	5,089,345	5,094,805	5,233,506	5,233,507
GC content (%)	57.67	57.67	59.83	59.83
Complete BUSCO (%)	98.65	98.65	99.32	99.32
**Features**				
Coding sequences (CDS)	4940	4942	5063	5065
Hypothetical	541	541	833	836
tRNA	71	74	76	75
tmRNA	1	1	1	1
rRNA	3	3	2	2
ncRNA	67	67	45	45
ncRNA Regions	48	53	50	50
sORF	2	2	4	4
oriC	2	2	2	2
Prophage regions	3	3	6	6
HGT regions	123	121	93	87
**Genes related to:**				
IE	23	25	22	21
RRR	121	121	94	94
P	73	73	110	109
STD	32	31	38	38
T	51	52	52	51
**Plasmid**				
Plasmid contigs	25	26	3	8
Combined length (bp)	292,234	291,521	101,176	118,671
Recovered plasmids	7	7	1	4
**Type**				
Conjugative	2	2	1	1
Mobilizable	2	2	0	2
Non-mobilizable	2	2	0	1
Discarded (AD092)	1	1	0	0

No CRISPR/Cas systems detected. Abbreviations: WGS, whole-genome sequencing. SRA, Sequence Read Archive. MLST, multilocus sequence typing. ST, sequence type. BUSCO, Benchmarking Universal Single-Copy Orthologs. IE, integration/excision. RRR, replication/recombination/repair. P, phage. STD, stability/transfer/defense (STD). T, transfer.

**Table 2 microorganisms-13-02286-t002:** Comparative analysis of conjugative, nonmobilizable, and mobilizable plasmids in CrKp and CrSm isolates.

Feature																	
Plasmid type	Conjugative						Nonmobilizable					Mobilizable					
Plasmid name	AA275		AA002				AB595			AA119		AA531			AB042		
Specie	*CrKp*	*CrKp*	*CrSm*	*CrSm*	*CrKp*	*CrKp*	*CrKp*	*CrSm*	*CrKp*	*CrKp*	*CrKp*	*CrSm*	*CrKp*	*CrKp*	*CrKp*	*CrKp*	*CrSm*
Patient	(2)	(1)	(2)	(3)	(2)	(1)	(1)	(3)	(2)	(1)	(2)	(3)	(1)	(2)	(1)	(2)	(3)
Replicon	IncFIB(K), IncFII(K)		IncM1				Col(pHAD28)			Col(pHAD28), Col440I		Col(pHAD28)					
Contigs count	15	14	3	4	5	5	2	2	2	1	1	1	1	1	1	1	1
Length (bp)	186,124	183,738	101,176	96,842	81,542	80,929	16,523	15,638	12,811	3674	3674	3115	3106	3106	3085	3085	3076
GC (%)	53.76	53.72	53.98	53.76	53.68	54.16	54.19	56.82	57.09	46.43	46.43	56.28	56.12	56.12	47.29	47.29	47.11
CDS	245	241	139	133	105	108	22	26	15	9	10	4	5	5	6	6	4
Hypothetical	97	95	89	87	73	77	7	12	4	7	8	3	4	4	4	4	2
HGT regions	5	5	3	3	3	2	-	-	-	-		-	-	-	-	-	-
Prophage regions	1	1	1	1	1	-	-	-	-	-		-	-	-	-	-	-
Prophage genes	4	4	5	5	5	-	-	-	-	-		-	-	-	-	-	-
**Genes related to:**																	
IE	45	43	22	20	11	11	4	5	2	-	-	-	-	-	-	-	-
RRR	9	9	9	7	7	7	-	2	-	1	1	-	-	-	-	-	-
P	1	1	2	2	1	1	-	-	-	-	-	-	-	-	-	-	-
STD	5	5	5	5	5	5	1	1	1	2	2	-	-	-	-	-	-
T	12	12	18	18	18	1	1	1	1	-	-	2	2	2	-	-	-

Abbreviations: CrKp, carbapenem-resistant *K. pneumoniae*. CrSm, carbapenem-resistant *S. marcescens.* IE, integration/excision. RRR, replication/recombination/repair. P, phage. STD, stability/transfer/defense (STD). T, transfer.

## Data Availability

The data presented in this study are openly available in NCBI at https://www.ncbi.nlm.nih.gov/search/all/?term=PRJNA1051122, reference number PRJNA1051122 (accessed on 14 August 2024).

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
