# Peer review of "Genomic Characterization of Carbapenem-Resistant Klebsiella pneumoniae ST1440 and Serratia marcescens Isolates from a COVID-19 ICU Outbreak in Ecuador"

_microorganisms, 2025, doi:10.3390/microorganisms13102286_

Round 1
Reviewer 1 Report
Comments and Suggestions for Authors
This manuscript describes investigations of two carbapenem-resistant K. pneumoniae and two carbapenem-resistant S. marcescens strains from a hospital outbreak. Genomic analysis was performed on all four strains. The topic of this manuscript is an interesting issue, however, some parts of this manuscript are unclear.
Comments
1. The title is very general. A more specific title should be added. I suggest this: Genomic analysis of carbapenem resistant Klebsiella pneumoniae ST1440 and carbapenem resistant Serratia marcescens from an outbreak in Ecuador
2. The introduction is very short and a more detailed description is needed about antibiotic resistance mechanisms, that are commonly present in Enterobacterales strains. Beta-lactamases, Amber classification, fluoroquinolone resistance determinants, aminoglycoside resistance genes should be described.
3. Infections caused by K. pneumoniae and S. marcescens should be also presented in the introduction.
4. In this study four strains were analyzed. What were the inclusion criteria? When were these strains isolated (year, month) ? These should be added to the materials and methods part.
5. I suggest to make a separate table about antibiotic resistance genes of all four analyzed strains.
6. My other suggestion would be to enlarge the part of figure 4, where the clusters are presented to get to a better view on data.
Author Response
Comment #1: The title is very general. A more specific title should be added. I suggest this: Genomic analysis of carbapenem resistant Klebsiella pneumoniae ST1440 and carbapenem resistant Serratia marcescens from an outbreak in Ecuador
Response: We appreciate the reviewer's suggestion and have modified the title accordingly to provide greater specificity regarding the bacterial species, resistance profile, and geographic location of the study.
Comment #2: The introduction is very short and a more detailed description is needed about antibiotic resistance mechanisms, that are commonly present in Enterobacterales strains. Beta-lactamases, Amber classification, fluoroquinolone resistance determinants, aminoglycoside resistance genes should be described.
Comment #3: Infections caused by K. pneumoniae and S. marcescens should be also presented in the introduction.
Response: We have expanded the introduction to include detailed descriptions of antibiotic resistance mechanisms in Enterobacterales, including beta-lactamases and their Ambler classification, fluoroquinolone resistance determinants, and aminoglycoside resistance genes. Additionally, we have added sections describing infections caused by K. pneumoniae and S. marcescens to provide better clinical context.
Comment #4: In this study four strains were analyzed. What were the inclusion criteria? When were these strains isolated (year, month)? These should be added to the materials and methods part.
Response: We have added the requested information regarding inclusion criteria and isolation dates (year and month) for all four analyzed strains in the Materials and Methods section.
Comment #5: I suggest to make a separate table about antibiotic resistance genes of all four analyzed strains.
Response: The comprehensive antibiotic resistance gene analysis results from staramr have been included as Supplementary File 1, providing detailed information for all four analyzed strains.
Comment #6: My other suggestion would be to enlarge the part of figure 4, where the clusters are presented to get to a better view on data.
Response: Figure 4 is already presented at maximum possible resolution. The detailed composition and characteristics of each cluster are thoroughly described in section 3.4 Clusters to complement the visual representation.

Reviewer 2 Report
Comments and Suggestions for Authors
Dear editor and authors
Concerning the case study, Impact of the COVID-19 Pandemic on Plasmid-Mediated antimicrobial Resistance: A Case Study of an ICU Outbreak in Third-Level Hospital
The MS is well written with representative data and conclusion just as this represent a case study more data about the case description, treatment and intervention need to be added,
Some minors such as all names of microbes need to be italic
Author Response
Comment #1: The MS is well written with representative data and conclusion just as this represent a case study more data about the case description, treatment and intervention need to be added. Some minors such as all names of microbes need to be italic
Response: We have added comprehensive case description details including treatment protocols and intervention measures implemented during the outbreak. Additionally, we have corrected the formatting to ensure all microbial names are properly italicized throughout the MS.

Reviewer 3 Report
Comments and Suggestions for Authors
The manuscript “Impact of the COVID-19 Pandemic on Plasmid-Mediated Antimicrobial Resistance: A Case Study of an ICU Outbreak in Third-Level Hospital” by Tisalema-Guanopatín et al. investigated carbapenem-resistant strains of Klebsiella pneumoniae ST1440 and Serratia marcescens isolated during an outbreak in the Intensive Care Unit (ICU) of a tertiary hospital in Quito, Ecuador. The findings revealed novel clusters of plasmid-mediated resistance genes, including efflux pump systems, AMR genes, and genes conferring heavy metal resistance, suggesting that coselective pressures during the pandemic contributed to the evolution and spread of these strains. The study emphasizes the critical need for improved infection surveillance and control, especially in resource-limited areas, to combat the growing threat of antimicrobial resistance.
While the article presented holds promise, aspects of it would benefit from additional refinement to enhance their robustness and rigor. Specific areas for improvement are outlined below:
Abstract:
There are many isolated sentences that could be linked together using connectors. I recommend that authors fix this.
Introduction:
Line 57: “throughhorizontal”. Separate these words.
Line 60: “byan”. Separate these words.
Lines 74-75: “However, there is a notable scarcity of research focused on Latin America and Ecuador”. I recommend that authors include a paragraph summarizing the limited studies on Enterobacteriaceae in Ecuador.
Methodology:
Since this is a study that will be relevant to genomic surveillance in Ecuador, I recommend that the authors include a methodology section detailing how DNA extraction, DNA quality and integrity assessment, sequencing library preparation, and sequencing were performed. This would ensure the reproducibility and applicability of this methodology in future studies in Ecuador.
Line 80: “Samples Collection”. Authors should specify the time of sample collection within the context of the pandemic. Additionally, including relevant patient characteristics would be beneficial. It would also be important to include the symptoms of the patients from whom the bacteria were isolated.
Line 84: “Microbiology Techniques”. I recommend specifying the incubation conditions as well as how the taxonomic identification of the bacteria was carried out at the phenotypic and genotypic levels..
Line 98: “Bioinformatic Analysis”. To enhance the clarity and reproducibility of your study, I recommend adding a paragraph to the methodology section. This paragraph should detail the rationale behind the sequence of bioinformatics analyses performed. Additionally, please include the specific parameters utilized in the programs, particularly those for gene and plasmid detection or identification (e.g., percentage of identity or coverage). I also recommend specifying which databases were used by the programs used. Providing this information, similar to the details given for DNA extraction and sequencing, is crucial for validating your results and enabling other researchers to reproduce your work.
I also recommend that authors confirm the taxonomic identification of each bacterium using its genome sequence. A phylogenetic analysis or Average Nucleotide Identity (ANI) could be employed to identify related strains.
Comments on the Quality of English Language
The manuscript requires revision by a native language specialist to address grammatical errors and improve paragraph structure.
Author Response
Comment #1: Abstract. There are many isolated sentences that could be linked together using connectors. I recommend that authors fix this.
Comment #2: Introduction: Line 57: “throughhorizontal”. Separate these words.
Comment #3: Line 60: “byan”. Separate these words.
Response: We have revised the abstract to improve flow by incorporating appropriate connectors between sentences for better readability. Additionally, we have corrected the typographical errors by separating "throughhorizontal" to "through horizontal" (line 57) and "byan" to "by an" (line 60).
Comment #4: Lines 74-75: “However, there is a notable scarcity of research focused on Latin America and Ecuador”. I recommend that authors include a paragraph summarizing the limited studies on Enterobacteriaceae in Ecuador.
Response: As suggested, we expanded the introduction to briefly summarize existing studies on carbapenem-resistant Enterobacterales in Ecuador, highlighting the lack of genomic analyses and the need for local surveillance data.
Comment #5: Methodology: Since this is a study that will be relevant to genomic surveillance in Ecuador, I recommend that the authors include a methodology section detailing how DNA extraction, DNA quality and integrity assessment, sequencing library preparation, and sequencing were performed. This would ensure the reproducibility and applicability of this methodology in future studies in Ecuador.
Response: We thank the reviewer for this important suggestion. In response, we have added a detailed subsection in the methodology describing the protocols used for DNA extraction, quality and integrity assessment, library preparation, and sequencing. This information will support future genomic surveillance studies in Ecuador by ensuring reproducibility and transparency of our workflow.
Comment # 6: Line 80: “Samples Collection”. Authors should specify the time of sample collection within the context of the pandemic. Additionally, including relevant patient characteristics would be beneficial. It would also be important to include the symptoms of the patients from whom the bacteria were isolated.
Response: We agree with the reviewer and have revised the "Sample Collection" section to indicate that all isolates were obtained between August and September 2021, during the COVID-19 pandemic. We have also included basic clinical details of the patients to provide better context for the strains and infection dynamics.
Comment # 7: Line 84: “Microbiology Techniques”. I recommend specifying the incubation conditions as well as how the taxonomic identification of the bacteria was carried out at the phenotypic and genotypic levels.
Response: We appreciate this recommendation. The revised “Microbiology Techniques” section now specifies the incubation conditions used for bacterial culture (temperature, time, atmosphere), as well as the methods used for phenotypic identification (biochemical profiling with VITEK®2) and genotypic confirmation through whole genome sequencing and ANI analysis via TYGS.
Comment # 8: Line 98: “Bioinformatic Analysis”. To enhance the clarity and reproducibility of your study, I recommend adding a paragraph to the methodology section. This paragraph should detail the rationale behind the sequence of bioinformatics analyses performed. Additionally, please include the specific parameters utilized in the programs, particularly those for gene and plasmid detection or identification (e.g., percentage of identity or coverage). I also recommend specifying which databases were used by the programs used. Providing this information, similar to the details given for DNA extraction and sequencing, is crucial for validating your results and enabling other researchers to reproduce your work.
Response: Thank you for this valuable observation. We have added a new paragraph in the “Bioinformatic Analysis” section explaining the rationale and sequence of the analyses performed. This includes the tools, their versions, the databases used (e.g., CARD, ResFinder, PLSDB), and the thresholds applied for gene and plasmid detection (e.g., ≥90% identity and ≥60% coverage). We also refer readers to Supplementary Table S1, which compiles all software tools used with version numbers and references.
Comment # 9: I also recommend that authors confirm the taxonomic identification of each bacterium using its genome sequence. A phylogenetic analysis or Average Nucleotide Identity (ANI) could be employed to identify related strains.
Response: We thank the reviewer for this suggestion. Taxonomic confirmation of all isolates was performed using genome-based ANI analysis through the TYGS platform. We have updated the MS accordingly, including this information in both the results and methods sections.

Round 2
Reviewer 3 Report
Comments and Suggestions for Authors
I have reviewed the resubmission of the manuscript entitled "Impact of the COVID-19 Pandemic on Plasmid- Mediated Antimicrobial Resistance: A Case Study of an ICU Outbreak in Third-level Hospital". The authors answered in a satisfactory way the points that I have addressed in the first review. Thus this new version of the manuscript can be accepted for publication.